# Structural basis for recognition of antihistamine drug by human histamine receptor

Xueqian Peng [1], Linlin Yang[2], Zixuan Liu [1], Siyi Lou [1], Shiliu Mei[1], Meiling Li[2], Zhong Chen [3] & Haitao Zhang [1,4] ✉

The histamine receptors belong to the G protein-coupled receptor (GPCR) superfamily, and play important roles in the regulation of histamine and other neurotransmitters in the central nervous system, as potential targets for the treatment of neurologic and psychiatric disorders. Here we report the crystal structure of human histamine receptor $H_3R$ bound to an antagonist PF-03654746 at 2.6 Å resolution. Combined with the computational and functional assays, our structure reveals binding modes of the antagonist and allosteric cholesterol. Molecular dynamic simulations and molecular docking of different antihistamines further elucidate the conserved ligand-binding modes. These findings are therefore expected to facilitate the structure-based design of novel antihistamines.

The biogenic amine histamine plays important pathophysiological roles in both the central nervous system (CNS) and periphery tissues, such as allergy, gastric acid secretion, neurotransmission, and immune response[1]. The action of histamine is mediated through four subtypes of G protein-coupled receptors (GPCRs), $H_1R$, $H_2R$, $H_3R$, and $H_4R$[2]. Antagonists of $H_1R$ and $H_2R$ have been clinically used for the treatment of allergies and gastric acid-related diseases, and the $H_3R$ inverse agonist Pitolisant (Wakix®) was approved for the treatment of narcolepsy[3]. While $H_4R$ antagonists are still in the clinical trials for their potential therapeutics in immune-related diseases[4]. Structures of $H_1R$ in complex with the agonist and antagonist have been determined[5,6], providing the molecular mechanisms for ligand recognition and facilitating the structure-based design of novel drugs targeting $H_1R$. However, the molecular mechanisms for ligand recognition with other histamine receptors were still elusive, due to the lacking of the $H_2R$, $H_3R$, and $H_4R$ structures.

$H_3R$ is expressed mainly in the brain and acts as an auto- or hetero-receptor in the histaminergic neurons[7]. As an auto-receptor,

$H_3R$ modulates the histamine release by the negative feedback[8]. While, as a hetero-receptor, $H_3R$ regulates the release of various neurotransmitters such as dopamine, γ-aminobutyric acid (GABA), and acetylcholine[9]. It was suggested that $H_3R$ was associated with several physiological progresses such as sleeping and wakefulness, learning and memory, feeding, and cerebral ischemia[10–12]. Therefore, $H_3R$ is a potential target for the treatment of neurologic and psychiatric disorders, such as sleep disorders, Parkinson's disease, schizophrenia, Alzheimer's disease, and cerebral ischemia[13,14]. The imidazole antagonist of $H_3R$ showed poor penetration through the blood–brain barrier and unwanted interactions with hepatic cytochrome P450[15]. Thus, great efforts have been devoted to the development of non-imidazole $H_3R$ antagonists[15]. Here we determine the crystal structure of human $H_3R$ bound to a non-imidazole antagonist PF-03654746 at 2.6 Å resolution. The structure, together with the computational and functional assays, reveals the critical interactions for the ligand binding, as well as the unexpected cholesterol binding at the allosteric site, which could accelerate the structure-based design of novel antihistamines.

[1]Hangzhou Institute of Innovative Medicine, Institute of Pharmacology and Toxicology, Zhejiang Province Key Laboratory of Anti-Cancer Drug Research, College of Pharmaceutical Sciences, Zhejiang University, 310058 Hangzhou, Zhejiang, China. [2]Department of Pharmacology, School of Basic Medical Sciences, Zhengzhou University, 450052 Zhengzhou, Henan, China. [3]Key Laboratory of Neuropharmacology and Translational Medicine of Zhejiang Province, College of Pharmaceutical Sciences, Zhejiang Chinese Medical University, 310053 Hangzhou, Zhejiang, China. [4]The Second Affiliated Hospital, Zhejiang University School of Medicine, 310009 Hangzhou, Zhejiang, China. ✉e-mail: haitaozhang@zju.edu.cn

## Results

### Overall structure of H3R

To obtain the stable human H3R proteins for structure determination, the flexible regions of the N-terminal residues 1–26, intracellular loop 3 (ICL3) residues 242–346, and C-terminal residues 433–445 were truncated, and a thermostabilized apocytochrome $b_{562}$RIL (BRIL) was inserted at the N-terminus. Additionally, a mutation of S121[3.39]K (superscript indicates residues numbers according to the Ballesteros–Weinstein scheme[16]) at the putative allosteric Na+ binding site was introduced to improve the homogeneity and thermostability of H3R as described in several GPCR structures determination[17–22] (Supplementary Fig. 1b, c). In our calcium mobilization assays, the crystallized construct of H3R with S121[3.39]K mutation could be activated by histamine with ~3-fold lower efficacy but inhibited by PF-03654746 with ~18-fold higher efficacy (Supplementary Fig. 2, Supplementary Table 1), which was in consistent with our results that the crystallized H3R-PF-03654746 proteins showed significantly improved homogeneity and thermostability (Supplementary Fig. 1). The crystal structure of H3R in complex with the antagonist PF-03654746 was determined at 2.6 Å resolution (Fig. 1, Supplementary Fig. 1, Supplementary Table 3).

The H3R structure consisted of the canonical seven transmembrane helical bundles (TMs1–7) connected by three extracellular loops (ECLs1–3) and three intracellular loops (ICLs1–3) with an amphipathic helix 8 (Fig. 1a). The ECL2 of H3R was stabilized by the conserved disulfide bridge between C107[3.25] and C188[ECL2], and the second disulfide bridge was found between C384[ECL3] and C388[ECL3] (Fig. 1a, b). Compared with the inactive H1R structure[5], the extracellular tips of TM6 and TM7 in H3R moved inwards by 2.3 and 3.5 Å, respectively (Fig. 1b). Additionally, the first section of ECL2-shifted towards TM3 by 11 Å and extended from the receptor core, otherwise the antagonist PF-03654746 would clash with ECL2 if it adopted a similar conformation to that in H1R (Fig. 1b). At the intracellular side, the TM6 of H3R showed an outward movement of

2.8 Å compared to the inactive H1R, whereas the active H1R showed the TM6 outward movement of 12 Å (Fig. 1c). Moreover, the ICL2 of H3R was found to form an additional helix (Fig. 1c, d). Notably, the Y[3.51] of D[3.49]–R[3.50]–Y[3.51] motif in H3R was substituted by F133[3.51], with the salt bridge formed between D131[3.49] and R132[3.50], which was a key feature of the inactive state of GPCRs[23] (Fig. 1d).

### PF-03654746 binding to H3R

In our H3R structure, PF-03654746 occupies a shallow pocket at the extracellular side, with clear densities for both the receptor and ligand (Fig. 2a). Although the orthosteric binding pocket of H3R is relatively shallow, an extended binding pocket (EBP) was found around TMs2/7 and ECL2 in H3R, compared to other aminergic receptors[24,25] (Fig. 2a). The ligand-binding pocket of H3R is constituted by the residues mainly from TMs2/3/6/7 and ECL2 (Fig. 2b). At the extracellular side, the carbonyl and N-ethyl-carboxamide moieties of PF-03654746 extends into the EBP by forming hydrophobic and hydrogen interactions with E395[7.36] and Y91[2.61], respectively (Fig. 2b). In our calcium mobilization assays, the E395[7.36]A mutant could fully abolish the PF-03654746 inhibition, while the Y91[2.61]A mutant could significantly decrease the PF-03654746 inhibition by ~46-fold (Supplementary Fig. 3a, Supplementary Table 1). Both Y[2.61] and E[7.36] are located in the minor pocket of aminergic GPCRs, which were shown to determine the ligand affinity and selectivity[26]. Additionally, the 3-fluoro-phenyl moiety of PF-03654746 formed hydrophobic interaction with F193[ECL2] (Fig. 2b). Mutating F193[ECL2] to alanine could completely abolish the PF-03654746 inhibition (Supplementary Fig. 3a, Supplementary Table 1). This phenylalanine on ECL2 was suggested to determine the ligand specificity among the aminergic receptors[27,28]. Moreover, the hydrophobic interaction with PF-03654746 is seen with Y374[6.51] (Fig. 2b). Mutagenesis of Y374[6.51]A could fully abolish the PF-03654746 inhibition

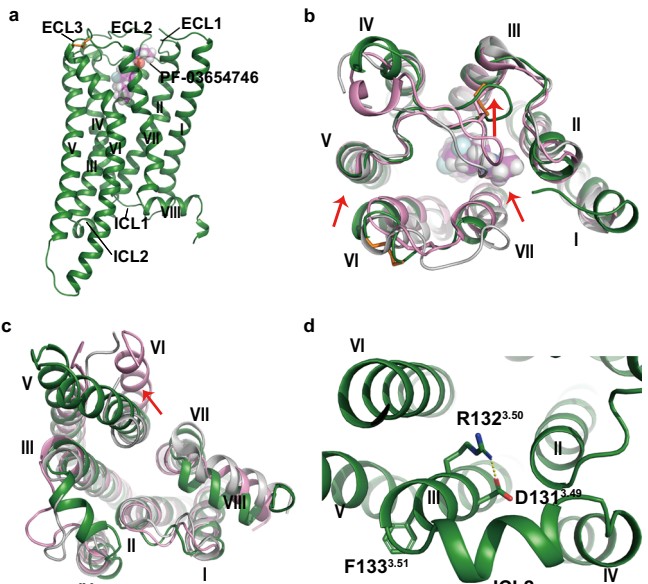

**Fig. 1 | Overall structure of H3R–PF-03654746 complex. a** Membrane view of H3R–PF-03654746 structure. H3R was shown in forest green ribbons. PF-03654746 was shown in a magenta sphere. The disulfide bond was shown as orange sticks. **b**, **c** Structural comparison of H3R (forest green) with inactive H1R (gray, PDB ID: 3RZE) and active H1R (pink, PDB ID: 7DFL) from extracellular view (**b**) and intracellular view (**c**). **d** Intracellular view showing a salt-bridge interaction (yellow dashed line) between D131[3.49] and R132[3.50]. The red arrows indicated movements of TMs5/6 and ECL2 in the H3R structure compared to the H1R inactive structure.

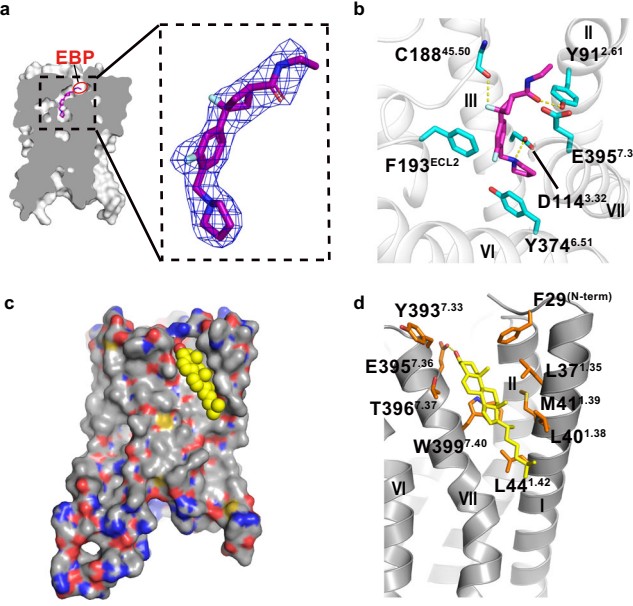

**Fig. 2 | Binding modes of PF-03654746 and cholesterol to H3R. a** Vertical cross section showing a shallow binding pocket in H3R. The extending binding pocket (EBP) of H3R-PF03654746 is shown in a red ellipse. $|2F_o|−|F_c|$ electron density map for the PF-03654746 contoured at 1.0σ. **b** Detailed interactions of PF-03654746 in the H3R ligand-binding pocket. H3R was shown in gray ribbons, with critical residues for ligand-binding as cyan sticks and PF-03654746 as magenta sticks. Hydrogen bonds were shown as yellow dashed lines. **c** Surface representation of cholesterol-binding site with cholesterol shown in yellow spheres. **d** Detailed interactions of cholesterol with H3R. Residues critical for cholesterol binding were shown as orange sticks and cholesterol was shown as yellow sticks. The hydrogen bond was shown as yellow dashed lines.

(Supplementary Fig. 3a, Supplementary Table 1). Notably, the fluorine atom of 3-fluoro-cyclobutane of PF-03654746 engages a hydrogen bond with C188$^{45.50}$, and the amine moiety of pyrrolidine of PF-03654746 forms a salt bridge with D114$^{3.32}$ at the bottom of the pocket (Fig. 2b), which is highly conserved in the aminergic receptors[28]. Surprisingly, both D114$^{3.32}$A and C188$^{45.50}$A mutations displayed similar PF-03654746 inhibition on the histamine-induced calcium mobilization compared to the wild-type (Supplementary Fig. 3a, Supplementary Table 1). However, the D114$^{3.32}$A and C188$^{45.50}$A mutants showed ~6-fold and ~4-fold reduction of histamine activation, indicating these two residues might be involved in the binding of both histamine and PF-03654746 (Supplementary Fig. 3a, Supplementary Table 1). Indeed, D$^{3.32}$ forms hydrogen bonds with histamine in H$_1$R[6].

## Cholesterol binding to H$_3$R

Cholesterol has been observed in many GPCR structures for its regulatory roles[29–33], at the classical cholesterol consensus motif (CCM)[34], as well as diverse binding sites[35–40]. In the adrenergic receptor β$_2$AR, two cholesterols bound at the CCM stabilizing the receptor conformation[34], while two other cholesterols were observed around helix 8 and TM1, modulating the β$_2$AR dimerization[39]. In the histamine receptors, the cholesterol-binding site was not identified previously. In our structure, the electron density of a cholesterol molecule is observed around TM1 and TM7 of H$_3$R (Fig. 2c). Cholesterol forms extensive hydrophobic interactions in the extrahelical pocket consisting of F29$^{N-term}$, L37$^{1.35}$, M41$^{1.39}$, L40$^{1.38}$, L44$^{1.42}$, T396$^{7.37}$, Y393$^{7.33}$, and W399$^{7.40}$. Especially, the β3-hydroxy head group of cholesterol interacts with E395$^{7.36}$ through hydrogen bonding (Fig. 2d). Notably, E395$^{7.36}$ also participates in the polar interactions with PF-03654736 (Fig. 2b). Our functional assays showed that mutating the negatively charged E395$^{7.36}$ to uncharged alanine or positively charged arginine had little effects on the histamine activation, while completely abolishing the PF-03654746 inhibition, indicating that cholesterol binding to E395$^{7.36}$ might not be critical for agonist binding and H$_3$R activation, but might potentially to affect antagonist binding and H$_3$R inhibition through an allosteric mode (Supplementary Fig. 3a, Supplementary Table 1).

To investigate the effects of cholesterol binding on H$_3$R, molecular dynamics (MD) simulations were performed on H$_3$R/PF-03654746 complex in the presence and absence of the crystal cholesterol molecule. Two systems, H$_3$R/PF-03654746/cholesterol (hereafter referred to as CHL) and H$_3$R/PF-03654746 (hereafter referred to as PF), were embedded in the palmitoyl oleoyl phosphatidylcholine (POPC) bilayer with a duration of 2000 ns, respectively, and each system was replicated to perform three independent simulations. A free-energy landscape was built to analyze the conformational changes in six 2-μs MD trajectories. RMSD$_{residues}$ and RMSD$_{PF}$, representing the root mean square deviations (RMSD) of orthosteric site residues and that of PF-03654746, respectively, were used as two collective variables of the landscape (Supplementary Fig. 4b). The small value of these parameters means the more approaching to the starting crystal conformation, while the larger value indicates obvious movements for both protein and PF-03654746.

The free-energy landscape showed three main minima corresponding to three states of the complexes: crystal-like state, state 2, and state 3 (Supplementary Fig. 4a). The crystal-like state contained snapshots from simulations CHL1, CHL2, and PF3 and displayed the smallest RMSD$_{PF}$ and RMSD$_{residues}$, representing the closest conformation to crystal structure. It is associated with the lowest free energy and is therefore the most stable. With larger RMSD$_{PF}$ and RMSD$_{residues}$, snapshots in simulation CHL3 formed state 2, and complexes from PF1 and PF2 fell into state 3. Both states were different from the crystal conformation and are characterized by higher free-energy values. In the crystal-like state, the PF-03654746-binding geometry was similar to that in the crystal structure, especially in the middle and bottom of the binding pocket (Supplementary Fig. 4a),

where salt bridges with D114$^{3.32}$ and hydrophobic interactions existed in every system. In the EBP, PF-03654746 was not that stable and adopted slightly different conformations, forming hydrogen bonds with Y91$^{2.61}$ in CHL1 and CHL2 systems or with Y94$^{2.64}$ in the PF3 system. Though PF-03654746 maintained the stable salt bridge with D114$^{3.32}$ in state 2, its conformation changed in the middle and external parts of the pocket and only occasionally interacted with A190$^{ECL2}$. For state 3, PF-03654746 totally lost its binding pose and rarely interacted with D114$^{3.32}$, resulting in a random orientation in each MD trajectory. It's noteworthy that the cholesterol molecule in CHL3 was not so stable as in CHL1 and CHL2 and eventually dissociated from its binding site at the TM1–TM7 interface (Supplementary Fig. 5a, c), so cholesterol-bound complexes only existed in simulations CHL1 and CHL2, and both of them were stabilized into the crystal-like conformations. Considering that one out of four cholesterol-unbound simulations also reproduced the crystal binding mode of PF-03654746, we came to the conclusion that cholesterol at the TM1–TM7 groove was not very stable and not the determining factor for complex stability, but bound cholesterol facilitated PF-03654746 present in the crystal pose at a higher frequency.

A significant phenomenon observed is that the conserved W399$^{7.40}$ played an essential role in stabilizing the cholesterol binding and ligand–H$_3$R interactions. W399$^{7.40}$ predominantly maintained the original rotameric state (RI-I, $\chi_1 \approx -80°$ and $\chi_2 \approx 100°$) in CHL1 and CHL2 (Supplementary Fig. 5b), and cholesterol resided stably in its site, forming a parallel π–π stacking with W399$^{7.40}$ (Supplementary Fig. 5a, c). But in the CHL3 simulation, the side chain of W399$^{7.40}$ flipped out of the TM1–TM7 cleft and pointed outward to the lipids at about 400 ns, resulting in a new rotamer conformation (RT-II, $\chi_1 \approx 175°$ and $\chi_2 \approx 100°$) (Supplementary Fig. 5b, d). The side chain flipping reduced π–π stacking and caused a big steric hindrance for the bound cholesterol. As a result, cholesterol gradually dissociated from the cleft (Supplementary Fig. 5a, c). Lacking the stabilization of cholesterol, the side chain of W399$^{7.40}$ turned to another conformation (RT-III) at about 1200 ns, and RMSD$_{PF}$ and RMSD$_{residues}$ in CHL3 greatly increased at the same time (Supplementary Fig. 4b). The observations above predicted that cholesterol regulated the complex dynamics by stabilizing W399$^{7.40}$ in RI-I state. To verify the role of W399$^{7.40}$ in ligand binding, we further analyzed the rotameric states of W399$^{7.40}$ in non-cholesterol system. As expected, W399$^{7.40}$ in PF1 and PF2 underwent a certain conformational change, while W399$^{7.40}$ of PF3 predominantly displayed RI-I state throughout the simulation, which should contribute to the stable conformation of H$_3$R/PF-03654746 complex obtained in this trajectory (Supplementary Fig. 5f).

To explore how W399$^{7.40}$ influenced the ligand binding, we examined its interactions with surrounding residues in the crystal structure. W399$^{7.40}$ formed T-shape π–π stackings with Y91$^{2.61}$, which was important for the PF-03654746 binding (Supplementary Fig. 5e). Indeed, mutation of W399$^{7.40}$A could completely abolish the PF-03654746 inhibition, while had little effects on the histamine activation (Supplementary Fig. 3a, Supplementary Table 1), indicating that cholesterol might affect the PF-03654746 binding mediated by the cholesterol–W399$^{7.40}$–Y91$^{2.61}$–PF-03654746 interactions. W399$^{7.40}$ and D114$^{3.32}$ are completely conserved, and W402$^{7.43}$ is highly conserved among monoamine receptors. Experiments have independently indicated the importance of W$^{7.40}$ for the ligand binding in several GPCRs[41,42]. Therefore, our study provided additional support for this idea and suggested a relevance between cholesterol and the W399$^{7.40}$–W402$^{7.43}$–Y91$^{2.61}$ motif.

More importantly, cholesterol facilitated rearrangements of the TM1–TM7 interface and stabilized a polar network of cholesterol–E395$^{7.36}$–R27$^{N-term}$. By making extensive hydrophobic contacts with the extrahelical part of TM1 and TM7, cholesterol joined TM1 and TM7 tightly like a 'glue' and promoted the formation of E395$^{7.36}$–R27$^{N-term}$ salt bridge (Supplementary Fig. 6a–c). Meanwhile,

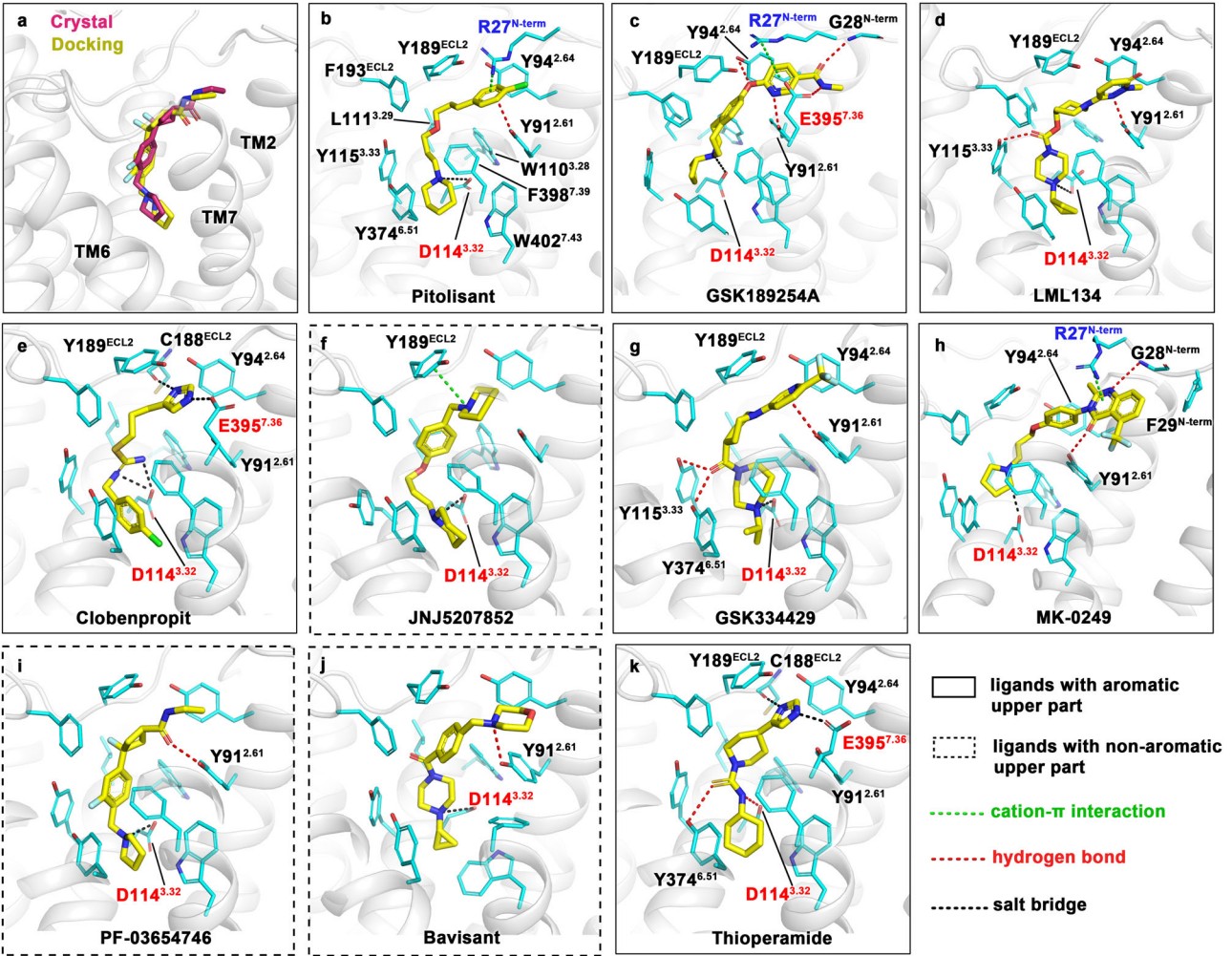

**Fig. 3 | Docking results of PF-03654746 and nine different H3R antagonists.**
**a** Superimposition of PF-03654746 in the crystal structure (magenta) and the docked pose (yellow). **b–k** Binding modes of 10 H3R antagonists. The docked ligands were depicted as yellow sticks and arranged according to their $K_i$ values from low to high as shown in Supplementary Table 2. Ligands with an aromatic moiety in the upper part of their binding poses were boxed with solid lines and ligands with non-aromatic upper parts were marked with dash lines. Interacting residues were presented as cyan sticks. In **b**, all residues involved in interactions were labeled. In **c–k**, only residues forming interactions with the external aromatic moiety or involved in polar contacts were labeled. Charged residues were marked in red (negative) or blue (positive). Polar interactions, including cation–π interactions, hydrogen bonds, and salt bridges, were represented by different colored dash lines, while π–π stackings were not shown.

the hydroxy of cholesterol established a stable hydrogen bond with the carboxyl group of E395[7.36] in our simulation, as indicated by the time dependences of their distance (Supplementary Fig. 5a). Hence, cholesterol–E395[7.36]–R27[N-term] polar network remained in CHL1 and CHL2, like in the crystal structure (Supplementary Fig. 6a). In the cholesterol-unbound simulations, only PF3 possessed the stable E395[7.36]–R27[N-term] salt bridge and similar compact conformation in TM1–TM7 interface. As for CHL3, PF1, and PF2, they showed declining stability of TM1 and TM7, as well as the E395[7.36]–R27[N-term] interaction (Supplementary Fig. 6b–e), consistent with their unstable complex states. Accordingly, the tight TM1–TM7–N-term contacts seemed to be favorable for ligand binding and cholesterol stabilized this receptor conformation through both hydrophobic and electrostatic interactions.

### Conserved binding modes of H₃R antagonists
To explore the binding modes of different H₃R antagonists, molecular docking studies were used to predict the binding conformations of other 9 H₃R antagonists (Fig. 3). PF-03654746 was first re-docked into the protein to verify the reliability of the docking simulation, which showed the RMSD < 3.0 Å with the solved crystal

structure (Fig. 3a). All ligands fit well in the binding pocket and all predicted docking scores were lower than −8.4 kcal/mol (Supplementary Table 4), which was inconsistent with the experimental $K_i$ values of these ligands (Fig. 3b–k)[1,13,43,44].

The docking results showed a common binding pose for all ligands. Apart from the conserved salt bridges with D114[3.32], docking studies revealed that favorable interactions between the aromatic upper part of ligands and residues in the EBP, as well as strong hydrophobic contacts at the bottom of the pocket, are of great importance for the ligand binding and efficacy. For all ligands, the downward heterocycle was in the hydrophobic pocket constituted by Y115[3.33], Y374[6.51], F398[7.39], and W402[7.43] (Fig. 3b). In the docked poses, conserved salt bridges were found between D114[3.32] and protonated nitrogen atoms at the bottom of the binding pocket for each ligand except Thioperamide. As Thioperamide did not get protonated at the equivalent position, it only formed a hydrogen bond with D114[3.32] (Fig. 3k), which might partly explain its worse inhibitive activity when compared with other antagonists (Supplementary Table 4). The middle part of the ligand was stabilized through hydrophobic interactions with L111[3.29], W110[3.28], F193[ECL2], and Y189[ECL2] (Fig. 3b), and the middle carbonyl group in GSK334429,

LML134, and the carbothioamide group in Thioperamide formed additional hydrogen bonds with Y374[6.51] and Y115[3.33] (Fig. 3d, g, k).

In the EBP, there were two general patterns of receptor–ligand interactions. Except for PF-03654746, JNJ5207852, and Bavisant, the other seven ligands all possess an aromatic moiety in the upper part of their binding poses, which could establish favorable π–π stacking interactions and OH/π hydrogen bonds with a cluster of aromatic residues in the EBP that involve Y91[2.61] and Y189[ECL2]. This was further validated by our functional assays that the Y91[2.61]A mutant significantly decreased the inhibition of GSK189254A and JNJ5207852 by ~25-fold and ~23-fold, respectively, and completely abolished the inhibition of Pitolisant (Supplementary Fig. 3b–d, Supplementary Table 2). While, the Y189[ECL2]A mutant decreased the inhibition of GSK189254A by ~88-fold and completely abolished the inhibition of Pitolisant (Supplementary Fig. 3b–d, Supplementary Table 2). On the other hand, ligands without aromatic moiety formed much fewer and weaker interactions in the EBP (Fig. 3f, i, j). Apparently, the extensive interactions benefit ligands binding and support the observation that most of the seven ligands with aromatic external moieties exert better inhibitive activity than ligands with non-aromatic groups (Fig. 3, Supplementary Table 4), highlighting the importance of aromatic rings in this part for H₃R antagonists. Additionally, with larger aromatic groups, GSK189254A and MK-0249 extended to reach the TM1–TM7 interface and even interacted with G28[N-term], E395[7.36], and F29[N-term] (Fig. 3c, h). The imidazole moiety in Thioperamide and Clobenpropit also formed polar interactions with C188[ECL2] and E395[7.36] in H₃R (Fig. 3e, k), in which the only non-conserved residue was R341[7.36] in H₄R (Supplementary Table 5), providing a structural basis for Thioperamide with similar affinity in H₃R and H₄R[1]. Indeed, the E395[7.36]A and E395[7.36]R mutants could fully abolish the inhibition of Thioperamide and Clobenpropit (Supplementary Fig 3e, f, Supplementary Table 2).

Further analysis suggested hydrophobic interactions at the bottom of the pocket play a role in ligand binding as well. Though JNJ5207852 and Bavisant showed similar contacts in the EBP through a protonated nitrogen atom (Fig. 3f, j), JNJ5207852 displays a much lower $K_i$ value (Supplementary Table 4), which may be the result of stronger hydrophobic packings made by the piperidine of JNJ5207852 than the cyclopropane of Bavisant. This could also be the reason why JNJ5207852 has better activity than GSK334429 and MK-0249 in spite that they formed more contacts in the EBP (Fig. 3f, g, h). It is the same in the case of Clobenpropit and Thioperamide. With identical interactions in the EBP, Clobenpropit not only established more powerful salt bridges with D114[3.32] as mentioned above but made more hydrophobic contacts through the fluorobenzene moiety at the bottom of the binding site (Fig. 3e, k).

Taken together, a combination of aromatic interactions in the EBP, salt bridges with D114[3.32] and hydrophobic patterns at the bottom of the pocket stabilized H₃R/antagonist complex. This exquisite binding feature rationalized the ability of Pitolisant, which possesses both a fluorobenzene group in the upper part and piperidine at the other end, to exhibit the best inhibitive activity among all ligands (Fig. 3b, Supplementary Table 4). In summary, the predicted poses of several H₃R antagonists demonstrate a conserved binding feature targeting H3R, which could facilitate the future structure-based drug design.

### Mechanism of H₃R antagonism

Structural comparison of our determined antagonist-bound H₃R structure with the inactive doxepin-bound H₁R[5] and active histamine-bound H₁R[6] structures provides an opportunity to visualize how the antagonist inhibits H₃R (Fig. 4a). A notable difference between H₁R and H₃R is the ligand-binding sites, where doxepin and histamine in H₁R bound deeply in the ligand-binding pocket, without interactions with the extracellular part (Fig. 4a). While, in H₃R, PF-03654746 occupies a shallow site near the extracellular part of the pocket, with only the

pyrrolidine adopting a similar position to the primary amino group of doxepin and histamine in H₁R (Fig. 4a). In the active structure of histamine-bound H₁R[6], three conserved residues D[3.32], T[3.37], and Y[6.51] form extensive hydrogen bonds with histamine and pushes TM6 towards TM3 for H₁R activation. In contrast, in the inactive structures of H₁R[5] and H₃R, neither the inverse agonist doxepin in H₁R nor the antagonist PF-03654746 in H₃R form hydrogen bonds with Y[6.51] (Fig. 4a). Y374[6.51] of H₃R forms hydrophobic interaction with PF-03654746 (Fig. 2b), and mutation of Y374[6.51]A could fully abolish the PF-03654746 inhibition, while showing little effects on histamine activation (Supplementary Fig. 3a, Supplementary Table 1), indicating Y374[6.51] might be critical for PF-03654746 binding but not histamine binding to H₃R. D114[3.32] might be an overlapping binding site for both histamine and PF-03654746 since D114[3.32]A mutant showed similar PF-03654746 inhibition on the histamine-induced calcium mobilization compared to the wild-type, but a ~6-fold reduction of histamine activation (Supplementary Fig. 3a, Supplementary Table 1). T119[3.37] in H₃R forms two intramolecular hydrogen bonds with E206[5.46], which is different from T112[3.37] in H₁R by forming hydrogen bonds with either doxepin or histamine (Fig. 4a). E206[5.46] of H₃R was suggested to form hydrogen bonds with the nitrogen atom in the imidazole ring of histamine and contribute to the binding of the selective H₃R agonist with a similar imidazole ring[13,45], indicating E206[5.46] might be critical for the H₃R activation.

Additionally, L401[7.42] forming hydrophobic interaction with PF-03654746 in H₃R corresponding to G457[7.42] in H₁R, which is likely to hinder the side chain of the toggle switch W371[6.48] in H₃R from forming a similar conformation in H₁R (Fig. 4a, Supplementary Table 4). In H₃R, the side chain of W371[6.48] is rotated ~90° and exhibits a perpendicular conformation relative to that in the H₁R structures (Fig. 4a, b). Consequently, the extracellular half of TM6 is pushed out by the outward displacement of W371[6.48] and Y374[6.51], thus expanding the ligand-binding pocket; contributing to the intracellular half of TM6 stabilizing an inactive state by forming the intramolecular hydrophobic interaction between W371[6.48] and F367[6.44] in the PIF motif. Indeed, the pocket volume of PF-03654746-bound H₃R (calculated by the CASTp 3.0 server[46]) was similar to that of the doxepin-bound inactive H₁R, but increased by ~3-fold in comparison with the histamine-bound active H₁R, which is in agreement with the expansion of the extracellular binding pocket in the inactive state of H₁R (Supplementary Fig. 7). Together with the intrahelical salt bridge observed between D[3.49] and R[3.50] in the DRY motif, and locked state of Y[7.53] in the NP[7.50]xxY[7.53] motif (Fig. 4d, e), these conformational changes resulted in an inactive state of H₃R in complex with PF-03654746 (Fig. 4a, Supplementary Table 4).

## Discussion

H₃R plays a crucial role in controlling the release of histamine and other neurotransmitters, and many studies have shown the therapeutic potentials of H₃R inverse agonists in CNS disorders[10], despite its complex pharmacology[13]. Drug discovery targeting H₃R was hampered by the lack of a three-dimensional structure to elucidate the molecular mechanisms for the ligand binding[45]. In this study, we reported a crystal structure of human H₃R in complex with an antagonist PF-03654746, which was developed for the treatment of CNS diseases. Our structure revealed a ligand-binding mode distinct from that of the antagonist-bound H₁R structure. Additionally, in combination with computational and functional assays, conserved binding modes of H₃R antagonists were identified, highlighting the importance of the residues in the EBP and the hydrophobic contacts at the bottom of the pocket for the ligand binding and efficacy. Especially, a cholesterol-binding site was identified next to the ligand-binding pocket, which might be targeted by the allosteric modulators. Our results are therefore expected to facilitate the structure-based novel antihistamine drug discovery targeting H₃R.

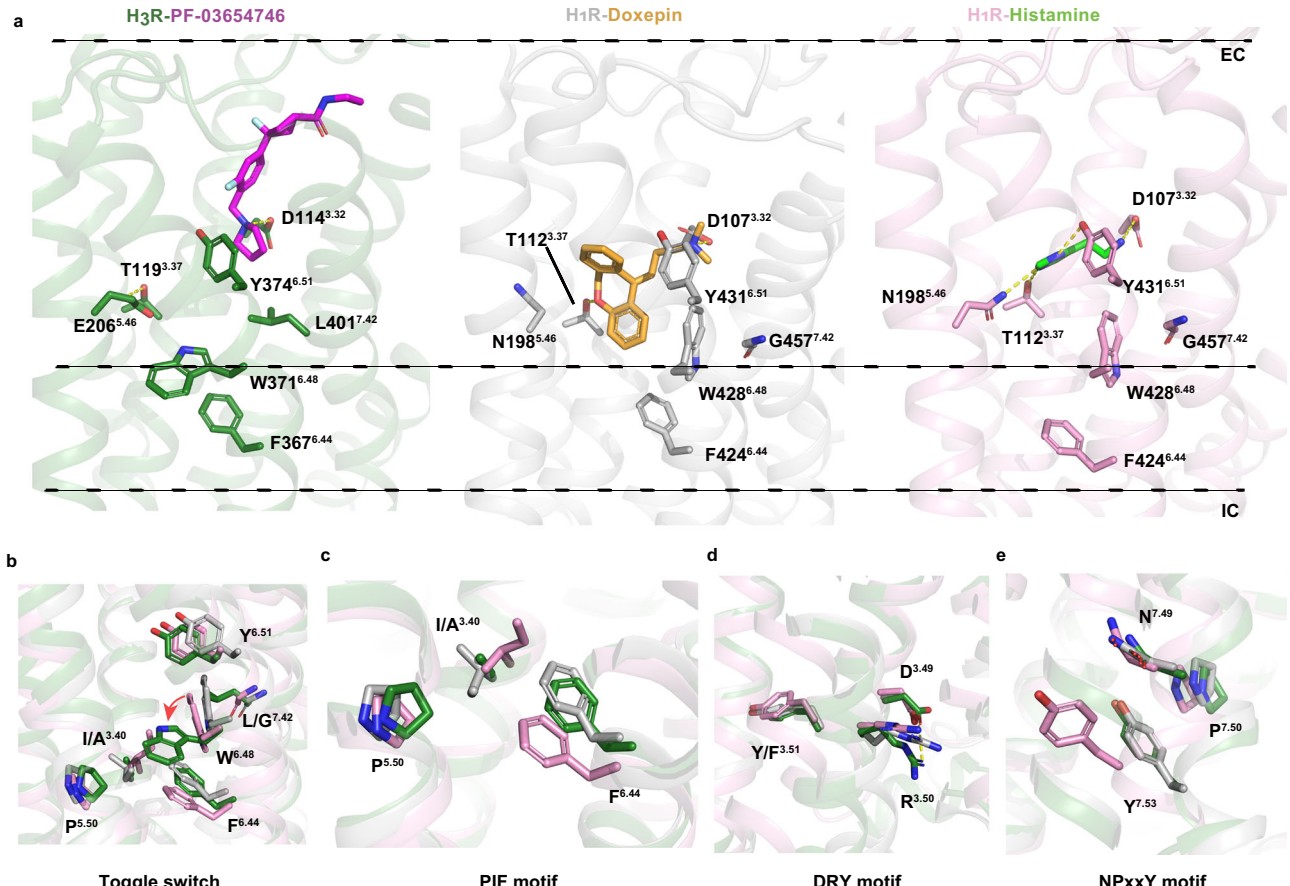

**Fig. 4 | Mechanism of H3R antagonism. a** Superpositions of the ligand-binding pockets of H3R–PF-03654746 (H3R in forest green, PF-03654746 in violet), H1R–Doxepin (PDB ID: 3RZE, H1R in gray, Doxepin in yellow), and H1R–Histamine (PDB ID: 7DFL, H1R in pink, Histamine in green) from the membrane view. Structural comparisons of the toggle switch (**b**), PIF motif (**c**), DRY motif (**d**), and NPxxY motif (**e**) with the same colors as **a**.

## Methods

### Protein engineering for structure determination

The codon-optimized human H3R gene was cloned into a modified pFastBac1 vector (Invitrogen) containing with N-terminal haemagglutinin (HA) signal sequence followed by a FLAG tag, a 10× His tag, and a tobacco etch virus (TEV) protease cleavage site. The H3R was modified by introducing S121$^{3.39}$K mutation to improve the thermostability and expression. To facilitate crystallization, N terminal residues 1–26 were replaced by the thermostabilized apocytochrome $b_{562}$RIL (BRIL) from *Escherichia coli* with mutations M7W, H102I, and R106L[47]. The ICL3 residues 242–346 and C terminal residues 433–445 were truncated.

### Protein expression and purification

The engineered H3R protein was expressed in *Spodoptera frugiperda* (*Sf9*) insect cells (Invitrogen) using the Bac-to-Bac Baculovirus Expression System. *Sf9* cells were infected at a density of 2–3 × 10⁶ cells per ml with a multiplicity of infection 5. Cells were harvested 48 h post-infection and stored at −80 °C until use.

Frozen biomass was thawed and disrupted by extensive washing in hypotonic buffer (10 mM HEPES, pH 7.5, 10 mM MgCl₂, 20 mM KCl) containing protease inhibitors (500 μM AEBSF, 1 μM E-64, 1 μM leupetain, 150 nM aprotinin) and high-osmotic buffer (10 mM HEPES, pH 7.5, 1.0 M NaCl, 10 mM MgCl₂, 20 mM KCl). Purified membranes were resuspended in the hypotonic buffer with the presence of 2 mg/mL iodoacetamide at 4 °C for 30 min, and then solubilized in 50 mM HEPES, pH 7.5, 800 mM NaCl, 0.5% (w/v) n-dodecyl-β-ᴅ-maltopyranoside (DDM, Anatrace), 0.1% (w/v) cholesterol hemisuccinate (CHS, Sigma-Aldrich), and 10% (v/v) glycerol for 3 h at 4 °C. After high-speed

centrifugation at 58,000×*g* for 1 h at 4 °C, the solubilized H3R proteins in the supernatants were incubated with TALON IMAC resin (TaKaRa) at 4 °C. After incubation overnight, the resin was then washed with 20 column volumes of washing buffer I (50 mM HEPES, pH 7.5, 800 mM NaCl, 10% (v/v) glycerol, 0.1% (w/v) lauryl maltose neopentyl glycol (LMNG, Anatrace), 0.01% (w/v) CHS, 20 mM imidazole), followed by 10 column volumes of wash buffer II (20 mM HEPES, pH 7.5, 500 mM NaCl, 5% (v/v) glycerol, 0.05% (w/v) LMNG, 0.005% (w/v) CHS, 40 mM imidazole). The protein was then eluted in 3 column volumes of elution buffer (10 mM HEPES, pH 7.5, 500 mM NaCl, 5% (v/v) glycerol, 0.01% (w/v) LMNG, 0.001% (w/v) CHS, 250 mM imidazole) and concentrated to 500 μL with a 100 kDa cutoff concentrator (Sartorius). Imidazole was removed by a PD MiniTrap G-25 column (GE Healthcare). Then, the sample was supplemented with 100 μM PF-03654746 and incubated with TEV protease overnight. The TEV protease, cleaved His-tag, and Flag-tag were removed by incubating with TALON IMAC resin (TaKaRa) at 4 °C for 2 h. The purified H3R–PF-03654746 complex protein was concentrated to ~40 mg/mL with a 100 kDa cutoff concentrator (Sartorius). The protein purity and monodispersity were tested by SDS–PAGE and analytical size-exclusion chromatography (aSEC).

### Lipidic cubic phase crystallization

Purified protein was reconstituted in LCP by mixing 40% of protein with 60% of lipid (monoolein and cholesterol, 9:1, w/w) using a syringe lipid mixer. Crystallization trials were performed on a Gryphon LCP robot (ArtRobbins) by dispensing 40 nL of protein-loaded LCP on 96-well glass sandwich plates and overlaying with 800 nL precipitant solution per well. Crystals appeared after 1 day and grew to full size

within 1 week in 0.1 M sodium cacodylate trihydrate, pH 6.4, 90 mM sodium citrate, 34% PEG400, and 0.005% dichloromethane. Crystals were collected directly from LCP using 50 μm micro-loops and flash-frozen in liquid nitrogen.

## Data collection and structure determination
The X-ray diffraction data of crystals were collected at the BL18U1 beamline of Shanghai Synchrotron Radiation Facility, using 20 μm × 20 μm beams for 0.8 s and 1° oscillation per frame with a Pilatus3 6M detector at a wavelength of 1.0000 Å. Diffraction data were processed with HKL3000[48]. Initial phase information was obtained by molecular replacement with CCP4[49] using $M_1R$[50] (PDB ID: 5CXV) and BRIL[51] (PDB ID: 1M6T) as search models. Refinement was performed with COOT[52] and Phenix[53] using $|2F_o|-|F_c|$ and $|F_o|-|F_c|$ maps. Pymol (http://www.pymol.org) was used to generate all the structural images in this manuscript.

## Molecular dynamics simulations
MD simulations on two systems ($H_3R$/PF-03654746/cholesterol system, $H_3R$/PF-03654746 system) were performed. Based on the crystal structure, we first built a complex model including $H_3R$, PF-03654746, and cholesterol. BRIL in the crystal structure was removed and the S121K mutation was mutated back to serine. To investigate the influence of cholesterol, we removed the cholesterol molecule to build a complex model only including $H_3R$ and PF-03654746. These models were separately placed into a 110 Å × 110 Å palmitoyl oleoyl phosphatidylcholine (POPC) bilayer and the lipids located within 1 Å of the receptor were removed. Both systems were solvated in a box (110 Å × 110 Å × 110 Å) with TIP3P water molecules and 0.15 M NaCl. Each system was replicated to perform three independent simulations and each of the three simulations was run up to 2-μs.

MD simulations were carried out with GROMACS 2020[54] with an isothermal–isobaric (NPT) ensemble and periodic boundary conditions. The CHARMM36-CMAP force field[55] was applied for protein, POPC phospholipids, cholesterol, ions, and water molecules. Ligand parameters were adapted from the CHARMM Generalized Force Field (CGenFF)[56,57]. For each system, stepwise energy minimizations were first performed to relieve unfavorable contacts with positional restraints imposed on i/protein, lipids, ligand, and cholesterol, ii/protein, ligand, and cholesterol, iii/mainchain atoms of protein, ligand, and cholesterol, iv/Cα atoms of protein, ligand, and cholesterol, v/no atoms. Subsequently, three parallel 50-ns equilibrations MD runs in the NPT ensemble were performed for each system with positional restraints applied in the same order as that in the energy minimization. During the equilibration, temperature and pressure were controlled using the v-rescale method[58] and the Berendsen barostatv[59], respectively. After equilibration, a 2-μs production run was carried out for each simulation. SETTLE constraints[60] and LINCS constraints[61] were applied to the hydrogen-involved covalent bonds in water molecules and in other molecules, respectively, and the time step was set to 2 fs. Electrostatic interactions were calculated with the particle-mesh Ewald (PME) algorithm[62] with a real-space cutoff of 1.0 nm. The temperature was maintained at 310 K using the v-rescale method[58] and the pressure was kept constant at 1 bar by semi-isotropic coupling to a Parrinello–Rahman barostat[63] with $\tau_p = 2.5$ ps and compressibility of $4.5 \times 10^{-5}$ bar. Analysis of simulation data was conducted using PyMOL (http://www.pymol.org), tools implemented in GROMACS 2020, and in-house scripts.

## Molecular docking
To investigate the interacting patterns between antagonists and $H_3R$, we performed flexible molecular docking studies using AutoDock 4[64]. The crystal structure of $H_3R$ reported here was used as the receptor and structures of 10 antagonists downloaded from the PubChem database were used as ligands. The receptor and ligands were respectively prepared by AutoDockTools to produce the corresponding low-energy three-dimensional conformation and the correct ionization state (pH 7.0). A 3D docking grid centered on PF-03654746 in the crystal structure was generated and residues around the pocket were treated as flexible. Then the processed antagonists were docked into the binding pocket of $H_3R$, outputting the top 10 conformations for each ligand. The most reliable binding poses were selected according to the interaction energy and visual inspection. All results were analyzed and visualized using PyMOL (http://www.pymol.org).

## Calcium mobilization assays
Calcium flux was performed as described in our previous studies. Briefly, CHO cells were co-transfected with wild-type or mutant $H_3R$ and $G_{qi5}$ using Lipofectamine 2000 according to the manufacturer's manual. Transfected cells were seeded into a 96-well flat clear bottom black plate with a density of 25,000 cells per well and cultured overnight. Subsequently, cells were loaded with calcium dye solution from Calcium 5 assay kit (Molecular Devices) in Hanks' balanced salt solution (20 mM HEPES, 2.5 mM probenecid in HBSS), and incubated at 37 °C for 45 min. Various concentrations of compounds were dispensed into the wells via a Flexstation III instrument (Molecular Devices). The intracellular calcium flux was detected immediately using the Flexstation III instrument (excitation at 485 nm, emission at 525 nm). Data were representative of three independent experiments and analyzed using GraphPad Prism 9.3.1.

## Reporting summary
Further information on research design is available in the Nature Research Reporting Summary linked to this article.

## Data availability
The data that support this study are available from the corresponding author upon reasonable request. The structural data generated in this study have been deposited in the Protein Data Bank (http://www.pdb.org/) under accession code 7F61. The other data generated in this study are provided in the Supplementary Information and Source Data file. Source data are provided with this paper.

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

## Acknowledgements

H.Z. is supported by the National Key R&D Program of China (2018YFA0508100), the National Natural Science Foundation of China (81722044, 91753115, 21778049, 81861148018), and the National Science and Technology Major Project of China (2018ZX09711002). We thank W.Q., Q.X., and other staff from the BL18U1 beamline of the National Facility for Protein Science in Shanghai (NFPS) at the Shanghai Synchrotron Radiation Facility, for assistance during data collection. We thank W.L. and M.L. from Shanghai Yuyao Biotech Ltd. for their assistance on the calcium mobilization assays.

## Author contributions

X.P. designed, expressed, purified, and crystallized the protein, collected the X-ray diffraction data. L.Y. and M.L. performed the computational assays. Z.L., S.L., and S.M. assisted in protein expression, purification, and crystallization. Z.C. supervised the functional assays. H.Z. conceived and supervised the project, and determined the structures. X.P., L.Y., Z.C., and H.Z. wrote the manuscript with input from all other authors.

## Competing interests

The authors declare no competing interests.
