## [Peer Review File · Nature Communications]

Structural basis for recognition of antihistamine drug by human histamine receptorREVIEWER COMMENTS

Reviewer #1 (Remarks to the Author):

In this manuscript, Peng et al. reported a crystal structure of an antagonist-bound H3R. The authors revealed the molecule recognition of PF-03654746 by the receptor and identified a cholesterol at an allosteric binding site. They also compared the antagonist-bound H3R with other members of the histamine receptor family, as well as other aminergic receptors. The manuscript is expected to provide structural clue for anti-histamine drug design, however, the paper suffers from the lack of mechanistic analysis and insightful information of receptor inhibition, and more importantly, the absence of experimental support.

The lack of the experimental support is the key issue of this paper. The paper uses plethora of in silico methods like MD simulations and docking to address the recognition of ligand and allosteric binding, however, there is not even a SINGLE experimental support of the paper's finding, conclusion. For instance, an experimental data of the S121K mutation used for crystallization is at least needed for illuminating the effect of this mutation on receptor. Also the recognition of PF-03654746 to receptor need to be confirmed in experiments, at least some of those key residues need to be validated in a mutagenesis study.

The focus of this paper should be how the antagonist inactivates receptor, not the allosteric binding of cholesterol. Cholesterol binding has been in many GPCRs, as well as other membrane proteins. Even with the extensive MD studies on the allosteric binding, the author still cannot draw an exclusive conclusion on the role of the cholesterol binding. A pocket size change has been seen in H1R agonist/antagonist binding, the authors should at least check the antagonist-bound pocket size of H3R with H1R.

The paper heavily depends on MD simulation and docking studies, need at least some experimental data to support the conclusions of MD and docking.

Figure S4 need show at least 3 three simulations to draw the conclusion.

A density of the antagonist PF-03654746 is at least needed in some of the figures (e.g. Fig 2b).

Reviewer #2 (Remarks to the Author):

Peng and colleagues report the crystal structure of the complex formed by histamine receptor H3 and antagonist PF-03654746. The generated coordinates are analyzed by computational means to test the stability of the antagonist binding at the orthosteric pocket as well as to investigate the effects of a cholesterol molecule binding at a newly characterized allosteric ligand binding site. Induced fit docking simulations are carried out to predict the binding modes of several known H3 binders, to retrospectively interpret their affinity, efficacy, and selectivity in light of the reported experimental coordinates. Last, the structure is thoroughly compared to previously reported crystal structures and/or AlphaFold models of other histamine receptor subtypes and other aminergic receptors.

The reported structure displays several peculiar if not unexpected features, which could not be simply extrapolated from other structures already reported. Hence, the findings described in the submitted manuscript are relevant for the scientific community.

My main concern is related to the molecular dynamics simulations. While the idea that

cholesterol affects binding at the orthosteric pocket is appealing, intuitive (given the proximity of the two binding sites) and very much in line with previous findings on the role of chol in GPCRs, it is here only partially supported by the reported results. In other words, the MD trajectory appears somewhat overinterpreted. I would prefer to see the results reported here confirmed even increasing the amount of sampling.

- a) By today's standards, even on a relatively large system such as this one, 1 microsecond can be considered a short amount of simulated time that only affords a limited amount of statistical power to the generated data
- b) Along the same lines, and more importantly, current best practices call for running multiple replicas of the same trajectory starting from different random distributions
- c) I strongly suggest letting both systems equilibrate longer in unrestrained runs before collecting data on interaction frequencies. In this way, the risk of an unfair comparison between a stable system generated from experimental coordinates and the other generated by substituting cholesterol with a POPC residue would be likely avoided
- d) Would an automatic analysis of the concerted motions along the trajectory (whether obtained by simple approaches e.g., essential dynamics, or by more complex methods for investigating contact networks) support the proposed idea that the interaction between the two binding sites is mainly mediated by Y91, E395 and W399?

The manuscript, which conveys a clear message, even if fairly well written, could still benefit if proofread by a native English speaker.

Reviewer #3 (Remarks to the Author):

The manuscript by Peng et al., describes the crystal structure of human histamine receptor H3R bound to an antagonist PF-03654746 at 2.6Å resolution. The receptor is an important drug target and the structure complexed with an H3R specific antagonist is a subject of interest of those developing drugs for neurologic and psychiatric disorders. This paper, however, very poorly presented the work. No biochemical data were presented to support their conclusions and they used may simulation results without experimental data to support them which is confusing and very difficult to judge if the discussions are credible. I do not recommend to publish the manuscript in the current form. For the further consideration at Nature Communications, significant improvement of the manuscript is necessary. They should address the following issues.

Major points:

- (1) PF-03654746 binding to H3R: The most import results of the paper is the binding mode of PF-03654746 but authors discuss little what makes PF-03654746 H3R specific. They should make the mutants of a few key residues to prove these residues are important for H3R selectivity. For the purpose, they should preferably make the mutants of other histamine receptors to show these residues are really determinants of the receptor specificity.
- (2) Cholesterol binding to H3R: They showed cholesterol binding stabilize the orthosteric ligand binding site by the MD simulation. They should do biochemical experiments to show if cholesterol increase the ligand affinity. Without this, it is very difficult to convince readers if the cholesterol binding is physiologically important.
- (3) Conserved binding mode of H3R antagonists: This section is all based on the results of docking simulations. This is only useful after careful evaluation of the ligand binding pocket as suggested in the point (1) above. Using all the compounds showed is very confusing. They should really highlight the key interaction between H3R and PF-03654746 for the ligand specificity and use the docking results to support this discussion.
- (4) Structural comparison with other histamine receptor subtypes: Again, this should be

done to discuss the H3R specificity of the ligand. They should compare the H3R structure with the inactive structure of H1R. The ligand binding pocket of active conformations of histamine receptors are largely different therefore should not be included for the comparison. The structures generated by the AlphaFold 2 should not be included. There is no quality control for these generated structures thus discussion is much more reliable without using them.

(5) Structural comparison with other aminergic receptors: There is little information in this comparison. I think the comparison to the inactive and active H1R (and H2R) structures are more informative which should be done in (4). This section should be removed or the discussion should be presented in the section of "Structural comparison with other histamine receptor subtypes." And the figure should be moved to the supplementary section.

Minor point:

(6) Supplementary Table I: Structure determination seems reasonable. However, authors should show some more standard statistics. For example, they should show R_{pim} and the last shell values of R_{work}/R_{free}.

REVIEWER COMMENTS

Reviewer #1 (Remarks to the Author):

In this manuscript, Peng et al. reported a crystal structure of an antagonist-bound H₃R. The authors revealed the molecule recognition of PF-03654746 by the receptor and identified a cholesterol at an allosteric binding site. They also compared the antagonist-bound H₃R with other members of the histamine receptor family, as well as other aminergic receptors. The manuscript is expected to provide structural clue for anti-histamine drug design, however, the paper suffers from the lack of mechanistic analysis and insightful information of receptor inhibition, and more importantly, the absence of experimental support.

Response: We thank the reviewer for the comments. We have rewritten a mechanistic analysis on the receptor inhibition, and we performed additional functional experiments to support our structural finding, as highlighted in the revised manuscript.

The lack of the experimental support is the key issue of this paper. The paper uses plethora of in silico methods like MD simulations and docking to address the recognition of ligand and allosteric binding, however, there is not even a SINGLE experimental support of the paper's finding, conclusion. For instance, an experimental data of the S121K mutation used for crystallization is at least needed for illuminating the effect of this mutation on receptor. Also the recognition of PF-03654746 to receptor need to be confirmed in experiments, at least some of those key residues need to be validated in a mutagenesis study.

Response: We thank the reviewer for the comments. We performed the functional experiments on the S121K mutation used for crystallization and the key residues mutations for the recognition of PF-03654746 to receptor, as described below and highlighted in the revised manuscript.

In lines 62-70: Additionally, a mutation of S121^{3,39}K (superscript indicates residues numbers according to the Ballesteros-Weinstein scheme¹⁶) at the putative allosteric Na⁺ binding site was introduced to improve the homogeneity and thermostability of H₃R as described in several GPCR structures determination^{17, 18, 19, 20, 21, 22} (Supplementary Fig. 1b, c). In our calcium mobilization assays, the crystallized construct of H₃R with S121^{3,39}K mutation could be activated by histamine with ~ 3-fold lower efficacy but inhibited by PF-03654746 with ~ 18-fold higher efficacy (Supplementary Fig. 2, Supplementary Table 1), which was in consistent with our results that the crystallized H₃R-PF-03654746 proteins showed significantly improved homogeneity and thermostability (Supplementary Fig. 1).

In lines 94-116: At the extracellular side, the carbonyl and N-ethyl-carboxamide moieties of PF-03654746 extended into the EBP by forming hydrophobic and hydrogen interactions with E395^{7,36} and Y91^{2,61}, respectively (Fig. 2b). In our calcium mobilization assays, the E395^{7,36}A mutant could fully abolish the PF-03654746 inhibition, while the Y91^{2,61}A mutant could significantly decrease the PF-03654746 inhibition by ~ 46-fold (Supplementary Fig. 3a, Supplementary Table. 1). Both Y^{2,61} and E^{7,36} were located in the minor pocket of aminergic GPCRs, which were indicated to determine the ligand affinity and selectivity²⁶. Additionally, the 3-fluoro-phenyl moiety of PF-03654746 formed hydrophobic interaction with F193^{ECL2} (Fig.

2b). Mutating F193^{ECL2} to alanine could completely abolish the PF-03654746 inhibition (Supplementary Fig. 3a, Supplementary Table. 1). This phenylalanine on ECL2 was suggested to determine the ligand specificity among the aminergic receptors^{27,28}. Moreover, the hydrophobic interaction with PF-03654746 were observed with Y374^{6.51} (Fig. 2b). Mutagenesis of Y374^{6.51}A could fully abolish the PF-03654746 inhibition (Supplementary Fig. 3a, Supplementary Table. 1). Notably, the fluorine atom of 3-fluoro-cyclobutane of PF-03654746 engaged a hydrogen bond with C188^{45.50}, and the amine moiety of pyrrolidine of PF-03654746 formed a salt bridge with D114^{3.32} at the bottom of pocket (Fig. 2b), which was highly conserved in the aminergic receptors²⁸. Surprisingly, both D114^{3.32}A and C188^{45.50}A mutations displayed similar PF-03654746 inhibition on the histamine-induced calcium mobilization compared to the wild-type (Supplementary Fig. 3a, Supplementary Table. 1). However, the D114^{3.32}A and C188^{45.50}A mutants showed ~ 6-fold and ~ 4-fold reduction of histamine activation, indicating these two residues might be involved in the binding of both histamine and PF-03654746 (Supplementary Fig. 3a, Supplementary Table. 1). Indeed, D^{3.32} formed hydrogen bonds with histamine in H₁R⁶.

The focus of this paper should be how the antagonist inactivates receptor, not the allosteric binding of cholesterol. Cholesterol binding has been in many GPCRs, as well as other membrane proteins. Even with the extensive MD studies on the allosteric binding, the author still cannot draw an exclusive conclusion on the role of the cholesterol binding. A pocket size change has been seen in H₁R agonist/antagonist binding, the authors should at least check the antagonist-bound pocket size of H₃R with H₁R.

Response: We thank the reviewer for the comments. In our revised manuscript, we analyzed the mechanism of H₃R inhibition and antagonist-bound pocket size of H₃R/H₁R, as described below and highlighted in the revised manuscript.

In lines 278-317: Mechanism of H₃R antagonism. Structural comparison of our determined antagonist-bound H₃R structure with the inactive doxepin-bound H₁R⁵ and active histamine-bound H₁R⁶ structures provided an opportunity to visualize how the antagonist inhibits H₃R (Fig. 4a). A notable difference between H₁R and H₃R is the ligand-binding sites, where doxepin and histamine in H₁R bound deeply in the ligand-binding pocket, without interactions with the extracellular part (Fig. 4a). While, in H₃R, PF-03654746 occupied a shallow site near the extracellular part of the pocket, with only the pyrrolidine adopting a similar position to the primary amino group of doxepin and histamine in H₁R (Fig. 4a). In the active structure of histamine-bound H₁R⁶, three conserved residues D^{3.32}, T^{3.37}, and Y^{6.51} formed extensive hydrogen bonds with histamine and pushed TM6 towards TM3 for H₁R activation. In contrast, in the inactive structures of H₁R⁵ and H₃R, neither the inverse agonist doxepin in H₁R nor the antagonist PF-03654746 in H₃R formed hydrogen bonds with Y^{6.51} (Fig. 4a). Y374^{6.51} of H₃R formed hydrophobic interaction with PF-03654746 (Fig. 2b), and mutation of Y374^{6.51}A could fully abolish the PF-03654746 inhibition, while showed little effects on the histamine activation (Supplementary Fig. 3a, Supplementary Table. 1), indicating Y374^{6.51} might be critical for PF-03654746 binding but not histamine binding to H₃R. D114^{3.32} might be an overlapped binding site for both histamine and PF-03654746, since D114^{3.32}A mutant showed similar PF-03654746 inhibition on the histamine-induced calcium mobilization compared to the wild-type, while ~ 6-fold reduction of histamine activation (Supplementary Fig. 3a, Supplementary Table. 1).

T119^{3.37} in H₃R formed two intramolecular hydrogen bonds with E206^{5.46}, which was different from T112^{3.37} in H₁R by forming hydrogen bonds with either doxepin or histamine (Fig. 4a). E206^{5.46} of H₃R was suggested to form hydrogen bonds with the nitrogen atom in the imidazole ring of histamine and contribute to the binding of the selective H₃R agonist with a similar imidazole ring^{13, 45}, indicating E206^{5.46} might be critical for the H₃R activation.

Additionally, L401^{7.42} forming hydrophobic interaction with PF-03654746 in H₃R corresponded G457^{7.42} in H₁R, which was likely to hinder the side chain of the toggle switch W371^{6.48} in H₃R from forming the similar conformation in H₁R (Fig. 4a, Supplementary Table 4). In H₃R, the side chain of W371^{6.48} rotated ~ 90° and exhibited a perpendicular conformation relative to that in the H₁R structures (Fig. 4a, b). Consequently, the extracellular half of TM6 was pushed out by the outward displacement of W371^{6.48} and Y374^{6.51}, thus expanding the ligand-binding pocket, contributing the intracellular half of TM6 to stabilize in an inactive state by forming the intramolecular hydrophobic interaction between W371^{6.48} and F367^{6.44} in the PIF motif. Indeed, the pocket volume of PF-03654746-bound H₃R (calculated by the CASTp 3.0 server⁴⁶) was similar to that of the doxepin-bound inactive H₁R, but increased by ~ 3-fold in comparison with the histamine-bound active H₁R, which was in agreement with the expansion of the extracellular binding pocket in the inactive state of H₁R (Supplementary Fig. 7). Together with the intrahelical salt bridge observed between D^{3.49} and R^{3.50} in the DRY motif, and locked state of Y^{7.53} in the NP^{7.50}xxY^{7.53} motif (Figure 4d, e), these conformational changes resulted in an inactive state of H₃R in complex with PF-03654746.

The paper heavily depends on MD simulation and docking studies, need at least some experimental data to support the conclusions of MD and docking.

Response: We thank the reviewer for the comments. We performed the functional experiments to support the conclusions of MD and docking, as described below and highlighted in the revised manuscript.

In lines 129-135: Our functional assays showed that mutating the negatively charged E395^{7.36} to uncharged alanine or positively charged arginine had little effects on the histamine activation, while completely abolished the PF-03654746 inhibition, indicating that the cholesterol binding to E395^{7.36} might not be critical for the agonist binding and H₃R activation, but might have potentials to affect the antagonist binding and H₃R inhibition in an allosteric mode (Supplementary Fig. 3a, Supplementary Table. 1).

In lines 192-196: Indeed, mutation of W399^{7.40}A could completely abolish the PF-03654746 inhibition, while had little effects on the histamine activation (Supplementary Fig. 3a, Supplementary Table. 1), indicating that cholesterol might affect the PF-03654746 binding mediated by the cholesterol-W399^{7.40}-Y91^{2.61}-PF-03654746 interactions.

In lines: 242-247: This was further validated by our functional assays that the Y91^{2.61}A mutant significantly decreased the inhibition of GSK189254A and JNJ5207852 by ~ 25-fold and ~ 23-fold, respectively, and completely abolished the inhibition of Pitolisant (Supplementary Fig. 3b-d, Supplementary Table 2). While, the Y189^{ECL2}A mutant decreased the inhibition of GSK189254A by ~ 88-fold and completely abolished the inhibition of Pitolisant (Supplementary Fig. 3b-d, Supplementary Table 2).

In lines 257-259: Indeed, the E395^{7.36}A and E395^{7.36}R mutants could fully abolish the

inhibition of Thioperamide and Clobenpropit (Supplementary Fig 3e, f, Supplementary Table 2).

Figure S4 need show at least 3 three simulations to draw the conclusion.

Response: We thank the reviewer for the comments. In our revised manuscript, we increased the amount of simulated time and sampling. Each system was replicated to perform three independent simulations and each of the three simulations was run up to 2- μ s.

A density of the antagonist PF-03654746 is at least needed in some of the figures (e.g. Fig 2b).

Response: We thank the reviewer for the comments. A density of the antagonist PF-03654746 is added in Fig2a.

Reviewer #2 (Remarks to the Author):

Peng and colleagues report the crystal structure of the complex formed by histamine receptor H3 and antagonist PF-03654746. The generated coordinates are analyzed by computational means to test the stability of the antagonist binding at the orthosteric pocket as well as to investigate the effects of a cholesterol molecule binding at a newly characterized allosteric ligand binding site. Induced fit docking simulations are carried out to predict the binding modes of several known H3 binders, to retrospectively interpret their affinity, efficacy, and selectivity in light of the reported experimental coordinates. Last, the structure is thoroughly compared to previously reported crystal structures and/or AlphaFold models of other histamine receptor subtypes and other aminergic receptors.

The reported structure displays several peculiar if not unexpected features, which could not be simply extrapolated from other structures already reported. Hence, the findings described in the submitted manuscript are relevant for the scientific community.

Response: We thank the reviewer for the comments.

My main concern is related to the molecular dynamics simulations. While the idea that cholesterol affects binding at the orthosteric pocket is appealing, intuitive (given the proximity of the two binding sites) and very much in line with previous findings on the role of chol in GPCRs, it is here only partially supported by the reported results. In other words, the MD trajectory appears somewhat overinterpreted. I would prefer to see the results reported here confirmed even increasing the amount of sampling.

Response: We thank the reviewer for the comments. In our revised manuscript, we have done further in-depth analysis of our running MD simulation and increase the amount of simulated time and sampling, which was rewritten the section of Cholesterol binding to H₃R in lines 118-214.

a) By today's standards, even on a relatively large system such as this one, 1 microsecond can

be considered a short amount of simulated time that only affords a limited amount of statistical power to the generated data

Response: We thank the reviewer for the comments. To better understand the effects of cholesterol binding on H3R, six molecular dynamics (MD) simulations were re-performed on two systems: H3R/PF-03654746 complex in the presence and absence of the crystal cholesterol molecule. The simulated time was extended to 2,000 ns to provide more powerful data and the complex reached a relatively stable conformation in each simulation, guaranteeing the subsequent analysis were carried out in stable systems (Supplementary Fig. 4b).

b) Along the same lines, and more importantly, current best practices call for running multiple replicas of the same trajectory starting from different random distributions

Response: We thank the reviewer for the comments. In our revised manuscript, each system was replicated to perform three independent simulations with different initial velocity distributions. As stated in the Method section, after stepwise energy minimizations, initial velocities were generated respectively in each of the three parallel 50-ns equilibration runs for both systems. During the equilibration, a random seed was used to generate velocity according to a Maxwell distribution for each simulation.

c) I strongly suggest letting both systems equilibrate longer in unrestrained runs before collecting data on interaction frequencies. In this way, the risk of an unfair comparison between a stable system generated from experimental coordinates and the other generated by substituting cholesterol with a POPC residue would be likely avoided

Response: We thank the reviewer for the comments. In our revised manuscript, three 2- μ s unrestrained production runs were carried out for each system after 50-ns equilibration. The longer simulation time enabled the substituted POPC molecule, surrounding residues as well as the whole system to adopt stable states (Supplementary Fig. 4b). Thus, analysis of simulation data was conducted in stable systems and comparisons between the crystal structure and complex conformations generated from the simulations became more reasonable.

d) Would an automatic analysis of the concerted motions along the trajectory (whether obtained by simple approaches e.g., essential dynamics, or by more complex methods for investigating contact networks) support the proposed idea that the interaction between the two binding sites is mainly mediated by Y91, E395 and W399?

Response: We thank the reviewer for the comments. In our revised manuscript, six 2- μ s MD trajectories provided tremendous information for conformational changes of the PF-03654746 binding pocket as well as cholesterol binding site. Comparative analysis suggested new ideas on how cholesterol allosterically regulate the ligand binding and/or receptor activation. We concluded cholesterol's role in three ways: (i) cholesterol regulated the complex dynamics by stabilizing W399^{7,40} in RI-I state ($\chi_1 \approx -80^\circ$ and $\chi_2 \approx 100^\circ$) (Supplementary Fig. 5), (ii) cholesterol had a direct influence on the T-shape π - π stackings in the W399^{7,40}-W402^{7,43}-Y91^{2,61} motif, which was important for ligand binding (Supplementary Fig. 5e), (iii) cholesterol facilitated the formation of a compact TM1-TM7 interface like a 'glue' through both hydrophobic and electrostatic interactions and stabilized a polar network of cholesterol-E395^{7,36}-R27^{N-term} (Supplementary Fig. 6).

The manuscript, which conveys a clear message, even if fairly well written, could still benefit if proofread by a native English speaker.

Response: We thank the reviewer for the comments. We have proofread by a native English speaker.

Reviewer #3 (Remarks to the Author):

The manuscript by Peng et al., describes the crystal structure of human histamine receptor H3R bound to an antagonist PF-03654746 at 2.6Å resolution. The receptor is an important drug target and the structure complexed with an H3R specific antagonist is a subject of interest of those developing drugs for neurologic and psychiatric disorders. This paper, however, very poorly presented the work. No biochemical data were presented to support their conclusions and they used may simulation results without experimental data to support them which is confusing and very difficult to judge if the discussions are credible. I do not recommend to publish the manuscript in the current form. For the further consideration at Nature Communications, significant improvement of the manuscript is necessary. They should address the following issues.

Response: We thank the reviewer for the comments.

Major points:

(1) PF-03654746 binding to H3R: The most important results of the paper is the binding mode of PF-03654746 but authors discuss little what makes PF-03654746 H3R specific. They should make the mutants of a few key residues to prove these residues are important for H3R selectivity. For the purpose, they should preferably make the mutants of other histamine receptors to show these residues are really determinants of the receptor specificity.

Response: We thank the reviewer for the comments. In our revised manuscript, we performed the functional experiments on the key residues' mutations for the binding mode of PF-03654746 to H3R, as described below and highlighted in the revised manuscript. And we added a new section of Mechanism of H₃R antagonism (lines 278-317).

In lines 94-116: At the extracellular side, the carbonyl and N-ethyl-carboxamide moieties of PF-03654746 extended into the EBP by forming hydrophobic and hydrogen interactions with E395^{7,36} and Y91^{2,61}, respectively (Fig. 2b). In our calcium mobilization assays, the E395^{7,36}A mutant could fully abolish the PF-03654746 inhibition, while the Y91^{2,61}A mutant could significantly decrease the PF-03654746 inhibition by ~ 46-fold (Supplementary Fig. 3a, Supplementary Table. 1). Both Y^{2,61} and E^{7,36} were located in the minor pocket of aminergic GPCRs, which were indicated to determine the ligand affinity and selectivity²⁶. Additionally, the 3-fluoro-phenyl moiety of PF-03654746 formed hydrophobic interaction with F193^{ECL2} (Fig. 2b). Mutating F193^{ECL2} to alanine could completely abolish the PF-03654746 inhibition (Supplementary Fig. 3a, Supplementary Table. 1). This phenylalanine on ECL2 was suggested to determine the ligand specificity among the aminergic receptors^{27,28}. Moreover, the hydrophobic interaction with PF-03654746 were observed with Y374^{6,51} (Fig. 2b). Mutagenesis

of Y374^{6.51}A could fully abolish the PF-03654746 inhibition (Supplementary Fig. 3a, Supplementary Table. 1). Notably, the fluorine atom of 3-fluoro-cyclobutane of PF-03654746 engaged a hydrogen bond with C188^{45.50}, and the amine moiety of pyrrolidine of PF-03654746 formed a salt bridge with D114^{3.32} at the bottom of pocket (Fig. 2b), which was highly conserved in the aminergic receptors²⁸. Surprisingly, both D114^{3.32}A and C188^{45.50}A mutations displayed similar PF-03654746 inhibition on the histamine-induced calcium mobilization compared to the wild-type (Supplementary Fig. 3a, Supplementary Table. 1). However, the D114^{3.32}A and C188^{45.50}A mutants showed ~ 6-fold and ~ 4-fold reduction of histamine activation, indicating these two residues might be involved in the binding of both histamine and PF-03654746 (Supplementary Fig. 3a, Supplementary Table. 1). Indeed, D^{3.32} formed hydrogen bonds with histamine in H₁R⁶.

(2) Cholesterol binding to H₃R: They showed cholesterol binding stabilize the orthosteric ligand binding site by the MD simulation. They should do biochemical experiments to show if cholesterol increase the ligand affinity. Without this, it is very difficult to convince readers if the cholesterol binding is physiologically important.

Response: We thank the reviewer for the comments. In our revised manuscript, we performed the functional experiments to support the MD conclusions on the cholesterol binding to H₃R, as described below and highlighted in the revised manuscript.

In lines 129-135: Our functional assays showed that mutating the negatively charged E395^{7.36} to uncharged alanine or positively charged arginine had little effects on the histamine activation, while completely abolished the PF-03654746 inhibition, indicating that the cholesterol binding to E395^{7.36} might not be critical for the agonist binding and H₃R activation, but might have potentials to affect the antagonist binding and H₃R inhibition in an allosteric mode (Supplementary Fig. 3a, Supplementary Table. 1).

In lines 192-196: Indeed, mutation of W399^{7.40}A could completely abolish the PF-03654746 inhibition, while had little effects on the histamine activation (Supplementary Fig. 3a, Supplementary Table. 1), indicating that cholesterol might affect the PF-03654746 binding mediated by the cholesterol-W399^{7.40}-Y91^{2.61}-PF-03654746 interactions.

(3) Conserved binding mode of H₃R antagonists: This section is all based on the results of docking simulations. This is only useful after careful evaluation of the ligand binding pocket as suggested in the point (1) above. Using all the compounds showed is very confusing. They should really highlight the key interaction between H₃R and PF-03654746 for the ligand specificity and use the docking results to support this discussion.

Response: We thank the reviewer for the comments. In our revised manuscript, we performed the functional experiments on the key residues' mutations for the binding modes of PF-03654746 and other compounds to H₃R, as described below and highlighted in the revised manuscript.

Lines 96-99: In our calcium mobilization assays, the E395^{7.36}A mutant could fully abolish the PF-03654746 inhibition, while the Y91^{2.61}A mutant could significantly decrease the PF-03654746 inhibition by ~ 46-fold (Supplementary Fig. 3a, Supplementary Table. 1).

Lines 102-103: Mutating F193^{ECL2} to alanine could completely abolish the PF-03654746 inhibition (Supplementary Fig. 3a, Supplementary Table. 1).

Lines 105-107: Mutagenesis of Y374^{6,51}A could fully abolish the PF-03654746 inhibition (Supplementary Fig. 3a, Supplementary Table. 1).

Lines 110-116: Surprisingly, both D114^{3,32}A and C188^{45,50}A mutations displayed similar PF-03654746 inhibition on the histamine-induced calcium mobilization compared to the wild-type (Supplementary Fig. 3a, Supplementary Table. 1). However, the D114^{3,32}A and C188^{45,50}A mutants showed ~ 6-fold and ~ 4-fold reduction of histamine activation, indicating these two residues might be involved in the binding of both histamine and PF-03654746 (Supplementary Fig. 3a, Supplementary Table. 1). Indeed, D^{3,32} formed hydrogen bonds with histamine in H₁R⁶.

Lines 242-247: This was further validated by our functional assays that the Y91^{2,61}A mutant significantly decreased the inhibition of GSK189254A and JNJ5207852 by ~ 25-fold and ~ 23-fold, respectively, and completely abolished the inhibition of Pitolisant (Supplementary Fig. 3b-d, Supplementary Table 2). While, the Y189^{ECL2}A mutant decreased the inhibition of GSK189254A by ~ 88-fold and completely abolished the inhibition of Pitolisant (Supplementary Fig. 3b-d, Supplementary Table 2).

Lines 257-259: Indeed, the E395^{7,36}A and E395^{7,36}R mutants could fully abolish the inhibition of Thioperamide and Clobenpropit (Supplementary Fig 3e, f, Supplementary Table 2).

(4) Structural comparison with other histamine receptor subtypes: Again, this should be done to discuss the H₃R specificity of the ligand. They should compare the H₃R structure with the inactive structure of H₁R. The ligand binding pocket of active conformations of histamine receptors are largely different therefore should not be included for the comparison. The structures generated by the AlphaFold 2 should not be included. There is no quality control for these generated structures thus discussion is much more reliable without using them.

Response: We thank the reviewer for the comments. In our revised manuscript, we have rewritten this part and analyzed the mechanism of H₃R inhibition in a new section of Mechanism of H₃R antagonism (lines 278-317).

(5) Structural comparison with other aminergic receptors: There is little information in this comparison. I think the comparison to the inactive and active H₁R (and H₂R) structures are more informative which should be done in (4). This section should be removed or the discussion should be presented in the section of “Structural comparison with other histamine receptor subtypes.” And the figure should be moved to the supplementary section.

Response: We thank the reviewer for the comments. We have removed the structural comparison with other aminergic receptors in our revised manuscript.

Minor point:

(6) Supplementary Table I: Structure determination seems reasonable. However, authors should show some more standard statistics. For example, they should show R_{pim} and the last shell values of R_{work}/R_{free} .

Response: We thank the reviewer for the comments. We have shown R_{pim} and the last shell values of R_{work}/R_{free} in the revised Table S3.

REVIEWERS' COMMENTS

Reviewer #1 (Remarks to the Author):

I am satisfied with the revision, the revision looks much better than the original one with the support of functional study, I therefore recommend an acceptance of the paper on Nature Communications.

Reviewer #2 (Remarks to the Author):

I am satisfied with the Authors' response to my comments and remarks.

Reviewer #3 (Remarks to the Author):

Authors have addressed my concerns properly. If the comments from other referees have been properly addressed, I am happy that the manuscript is accepted for publication.